# Changes in Cigarette Smoking and Vaping in Response to the COVID-19 Pandemic in the UK: Findings from Baseline and 12-Month Follow up of HEBECO Study

**DOI:** 10.3390/ijerph19020630

**Published:** 2022-01-06

**Authors:** Dimitra Kale, Olga Perski, Aleksandra Herbec, Emma Beard, Lion Shahab

**Affiliations:** 1Department of Behavioural Science and Health, University College London, London WC1E 7HB, UK; Olga.perski@ucl.ac.uk (O.P.); a.herbec@ucl.ac.uk (A.H.); e.beard@ucl.ac.uk (E.B.); lion.shahab@ucl.ac.uk (L.S.); 2SPECTRUM Research Consortium, Edinburgh EH8 9YL, UK; 3Department of Clinical, Educational and Health Psychology, University College London, London WC1E 6BT, UK

**Keywords:** smoking, vaping, cessation, COVID-19, UK, cohort study, socioeconomic factors

## Abstract

This study investigated UK adults’ changes in cigarette smoking and vaping during the COVID-19 pandemic and factors associated with any changes. Data were from an online longitudinal study. A self-selected sample (n = 332) of 228 smokers and 155 vapers (51 participants were both smokers and vapers) completed 5 surveys between April 2020 and June 2021. Participants self-reported data on sociodemographics, COVID-19-related, and smoking/vaping characteristics. During the 12 months of observations, among smokers, 45% self-reported a quit attempt (27.5% due to COVID-19-related reasons) since the onset of COVID-19 pandemic and the quit rate was 17.5%. At 12 months, 35.1% of continuing smokers (n = 174) reported smoking less and 37.9% the same, while 27.0% reported an increase in the number of cigarettes smoked/day. Among vapers, 25.0% self-reported a quit attempt (16.1% due to COVID-19-related reasons) and the quit rate was 18.1%. At 12 months, 47.7% of continuing vapers (n = 109) reported no change in the frequency of vaping/hour, while a similar proportion reported vaping less (27.5%) and more (24.8%). Motivation to quit smoking and being younger were associated with making a smoking quit attempt and smoking cessation. Being a cigarette smoker was associated with vaping cessation. Among a self-selected sample, COVID-19 stimulated more interest in reducing or quitting cigarette smoking than vaping.

## 1. Introduction

Given the known impact of cigarette smoking on respiratory disease and immune function [1], the onset of the coronavirus (COVID-19) pandemic raised concerns among public health professionals that smokers may be at a greater risk of COVID-19 infection, severe disease, and death [2]. While there is some evidence that current compared with never smokers admitted to hospital with COVID-19 are at increased risk of severe disease and death [3,4,5], other systematic reviews suggest that current compared with never smokers have a reduced risk of initial COVID-19 infection [6,7]. E-cigarette use (hereafter referred to as vaping) is associated with substantially reduced levels of measured carcinogens and toxins relative to cigarette smoking and thus are less harmful [8]. However, there is a similar public health concern that vaping may increase harm from COVID-19 [2], though evidence is lacking [9,10]. These inconclusive findings notwithstanding, cigarette smoking and vaping remain serious concerns during the COVID-19 pandemic regarding the impact of these behaviours on COVID-19 outcomes as well as the impact of COVID-19 on these behaviours.

Apart from understanding the direct health effects of COVID-19 on people who smoke and/or vape, characterising the pandemic’s impact on smoking and vaping behaviour is another important area of investigation. However, findings have thus far been equivocal and limited to the first few months of the pandemic. For instance, a nationally representative survey in the UK found that the first national lockdown (beginning of March 2020 to July 2020) was accompanied by an increase in motivation to quit smoking and the number of quit attempts [11]. Further, reductions in smoking behaviour have been found across several surveys in many countries. Within samples in the UK and US, a substantial proportion of smokers reported an increase in motivation to quit smoking (35%; [12]), increased quit attempts (12–23%; [12,13]), or an actual reduction in smoking frequency during the pandemic (28%; [12,14]). Although a non-trivial proportion of smokers have engaged in smoking cessation attempts during the pandemic, research also suggests that the majority of smokers did not change their behaviour, and in fact, that many smokers increased their smoking frequency [12,13,14]. This may be attributed to the fact that cessation attempts triggered by the pandemic were largely unaided [15]. Research in the UK suggests no change in downloads of a popular smoking cessation app during the initial months of the COVID-19 pandemic [16], though a study in the US from the same time period reported an increase in traffic on the Smokefree website and adult-focused digital intervention platforms in 2020 [17]. Likewise, research on vaping suggests that a proportion of vapers decreased their product use in the first months of the COVID-19 pandemic (10–24%; [10,14]), though some vapers reported an increase (24–40%; [10,14]), and half did not change their use (50%; [10]).

For some, boredom and restrictions in movement during lockdowns or other behavioural restrictions might have stimulated increased smoking and vaping [10,18], while for others, concerns about contracting COVID-19 and becoming severely ill might have motivated them to improve their health by stopping smoking and vaping [10,19]. For instance, perceived risk of severe infection from COVID-19 was found to be a positive predictor of motivation to quit smoking [20], and smokers who reported that COVID-19 was a greater risk to smokers than non-smokers also reported a reduction in their smoking behaviour in the first months of the pandemic [14]. In addition, smokers and vapers who had a direct experience with COVID-19, such as someone in their household testing positive, reported a stronger desire to quit smoking and/or vaping [19]. Irrespective of COVID-19-related reasons for quit attempts and quit rate, research suggests that quit attempts are linked with motivation to quit, while quit rate is linked with nicotine dependence [21,22]. 

Critically, although extant studies provide useful information on changes in smoking and vaping behaviour in response to the COVID-19 pandemic, they have been limited to the first few months of the pandemic, and they have largely been cross-sectional. Since COVID-19 restrictions are constantly changing and COVID-19 vaccines have been widely administered since the beginning of 2021 (at least in high-income countries), these may have changed attitudes and behaviours towards COVID-19. Longitudinal studies have reported that many health behaviours (i.e., physical activity [23] and dietary behaviour [24]) have changed dynamically over the course of the pandemic and in response to changing COVID-19 restrictions. As such, smoking and vaping behaviour may not have remained constant over the course of the COVID-19 pandemic.

This study uses data collected over 12 months to investigate the long-term effects of the COVID-19 pandemic on smoking and vaping in UK adults. Specifically, this study aimed to investigate how smoking and vaping have changed during the first year of the COVID-19 pandemic, to identify factors associated with any changes, and to explore whether COVID-19 has acted as a source of motivation for smokers and vapers to quit. The research questions were:

Among cigarette smokers and vapers at baseline (April–June 2020; covering the period of the first national COVID-19-related lockdown in the UK): 

RQ1. What proportion at a 12-month follow up (May–June 2021; ease of COVID-19-related restrictions in UK) reported having made a quit attempt and a successful quit attempt?

RQ2. What proportion at a 12-month follow up (May–June 2021) reported having changed their cigarette or vaping consumption?

RQ3. Which, if any sociodemographic, smoking/vaping, and COVID-19 related characteristics are associated with quit attempts, quit rate, and changes in consumption?

RQ4. What proportion self-reported COVID-19 related reasons for making a quit attempt? 

## 2. Materials and Methods

### 2.1. Study Design

Analysis of longitudinal data of a prospective online survey of adults residing in the UK; the HEalth BEhaviours during the COVID-19 pandemic (HEBECO) study (accessed 17 December 2021. https://osf.io/sbgru/). The study was approved by the Ethics Committee at the UCL Division of Psychology and Language Sciences (CEHP/2020/759). Baseline data collection occurred between April and June 2020, and follow-up surveys were administered at 1 month (FU1), 3 months (FU2), 6 months (FU3), and 12 months (FU4) from the baseline participation date. This analysis uses data from the baseline and 12-month follow up, apart from quit attempts, which were measured at each follow up.

### 2.2. Study Sample

This study included a self-selected sample of UK-based adult (18+ years) who were either smokers or vapers (some were both smokers and vapers, i.e., dual users) and completed the baseline survey of the HEBECO study between 23 April 2020 (initiation of recruitment) and 14 June 2020 inclusive (marking the end of the first national UK lockdown) and were successfully followed-up after 12 months (FU4; ease of COVID-19-related restrictions in the UK). A total of 2994 participants completed the baseline questionnaire of the HEBECO study, of whom 751 were smokers and vapers (556 smokers; 337 vapers; among the smokers and vapers there were 142 dual users), who were potentially eligible for this study. Of these eligible 751 participants at baseline, 332 (44.2%) were successfully followed-up at 12 months. Participants who were successfully followed-up at 12 months were significantly older (*p* < 0.001), more were of white ethnicity (*p* = 0.004), with post-16 education qualifications (*p* = 0.004), and non-smokers (*p* = 0.003) than those who did not complete the 12-month follow-up (Table 1).

Initial recruitment at baseline was online and involved sharing study invitations via multiple channels, including unpaid and paid advertisements on social media (e.g., Facebook, Twitter, Reddit), an email campaign across the network of UCL, other universities in the UK, Public Health England, Cancer Research UK, charities, and local authorities across the UK. The full recruitment strategy is available online (accessed 17 December 2021. https://osf.io/sbgru/). 

Participants gave their written consent prior to data collection. Data were captured and managed within the REDCap electronic data system [25,26]. Participants were followed up via email (except for participants who explicitly opted out), with up to three reminders to complete the survey sent at each follow up. Reasons for not completing the follow-up surveys were not assessed.

### 2.3. Measures

All measures were self-reported.

#### 2.3.1. Outcomes

Smoking status was assessed at baseline and at each follow up with the question ‘Which statement about tobacco use and cigarette smoking best describes you?’ [27], with the options (i) I smoke cigarettes (including hand-rolled) every day; (ii) I smoke cigarettes (including hand-rolled), but not every day; (iii) I do not smoke cigarettes at all, but I do smoke tobacco of some kind (e.g., pipe, cigar or shisha); (iv) I have stopped smoking completely in the last year; (v) I stopped smoking completely more than a year ago; and (vi) I have never smoked any cigarettes. Those who selected (i) or (ii) were classified as smokers and all the others as non-smokers. Participants who were smokers at baseline and non-smokers at the 12-month follow-up were considered as having quit smoking successfully since the COVID-19 pandemic. 

Smoking quit attempts (among smokers at baseline) assessed at baseline with the question ‘How many quit attempts have you made since COVID-19?’ with the option zero to any number, and at each of the four follow-up waves with the question ‘Have you tried to stop smoking for good in the [timeframe since the previous follow-up]?’ [27] with the option (i) yes or (ii) no. Participants who reported at least one quit attempt either at baseline or at the follow ups were categorised as having made a quit attempt since the onset of COVID-19 pandemic.

Vaping status assessed at baseline and each follow up with the question ‘Which statement about vaping (e-cigarette use) best describes you?’ (adapted from [27]), with the options (i) I vape or use e-cigarettes every day; (ii) I vape or use e-cigarettes but not every day; (iii) I stopped vaping or using e-cigarettes completely in the last year; (iv) I stopped vaping or using e-cigarettes completely more than a year ago; (v) I have never vaped or used e-cigarettes. Those who selected (i) or (ii) were classified as vapers and all the others as non-vapers. Participants who were vapers at baseline and non-vapers at the 12-month follow up were considered as having quit vaping successfully since the COVID-19 pandemic.

Vaping quit attempts (among vapers at baseline) assessed at baseline with the question ‘How many quit attempts have you made since COVID-19?’ with the option zero to any number, and at each of the four follow-up waves with the question ‘Have you tried to quit vaping for good in the [timeframe since the previous follow-up]?’ (adapted from [27]) with the option (i) yes or (ii) no. Participants who reported at least one quit attempt either at baseline or at the follow-ups were categorised as having made a quit attempt since the onset of the COVID-19 pandemic.

Changes in the number of cigarettes smoked per day (assessed among smokers at baseline who were also smokers at the 12-month follow up) was derived from a question asking about the number of cigarettes smoked per day (answer option 1–40+ cigarettes per day; [28]) at two time points: (i) at baseline and (ii) at the 12-month follow up. Three variables for change in smoking from baseline to the 12-month follow up were developed: (1) increased smoking, (2) decreased smoking, (3) no change.

Changes in frequency of vaping per hour (assessed among vapers at baseline who were also vapers at the 12-month follow up) was derived from a question asking about the number of times per hour of e-cigarette use (answer options: (i) less than once, (ii) once, (iii) up to 5 times, (iv) up to 10 times, (v) nearly all the time, (vi) don’t know, adapted from [28]) at two time points: (i) at baseline and (ii) at the 12-month follow-up. Three variables for changes in vaping from baseline to the 12-month follow-up were developed: (1) increased vaping, (2) decreased vaping, (3) no change. People who responded ‘don’t know’ were excluded from this analysis.

Reasons for making a smoking/vaping quit attempt (assessed among smokers and vapers at baseline who made a quit attempt) assessed at all waves and included: (1) Rules around social distancing/self-isolation due to COVID-19, (2) Children/parents moved back home due to COVID-19, (3) Money is tighter due to COVID-19, (4) Decided it was too expensive, (5) Health problems/concerns related to COVID-19, (6) Health problems/concerns unrelated to COVID-19, (7) Advice from a GP, (8) Government/Tv/radio/press advert, (9) Social campaign, (10) Being contacted by local NHS Stop Smoking Services, (11) Being faced with restrictions already before COVID-19, (12) I knew someone else who was stopping, (13) Seeing a health warning on a packet, (14) Something said by family/friends/children, (15) Improve fitness, (16) Other. Participants could select one or more reasons. Participants who selected at least one of reasons 1, 2, 3, or 5 were classified as motivated to quit smoking/vaping due to COVID-19-related reasons. 

#### 2.3.2. Predictors/Covariates

Predictors were assessed at baseline, unless otherwise stated. 

Sociodemographic characteristics included age (continuous in years), gender (female vs all other), education (post-16 qualification vs no post-16 qualification), ethnicity (any white ethnicity vs all other including ‘prefer not to say’), household income (≥£50,000 vs. <£50,000 GBP vs prefer not to say; ‘prefer not to say’ was categorised separately as almost 10% of participants selected this answer option), health conditions (no vs. yes including ‘prefer not to say’).

Smoking/vaping characteristics included motivation to quit smoking (among smokers at baseline)/vaping (among vapers at baseline), assessed with the question ‘Which of the following best describes you?’ (Motivation to Stop Scale; [29]) with the options ‘(i) I REALLY want to stop smoking/vaping and intend to in the next month, (ii) I REALLY want to stop smoking/vaping and intend to in the next 3 months, (iii) I want to stop smoking/vaping and hope to soon, (iv) I REALLY want to stop smoking/vaping but I don’t know when I will, (v) I want to stop smoking/vaping but haven’t thought about when, (vi) I think I should stop smoking/vaping but don’t really want to, (vii) I don’t want to stop smoking/vaping’. Those who selected (i–v) were considered motivated to quit smoking/vaping. 

Cigarette dependence (among smokers at baseline) was assessed with the Heaviness of Smoking Index (HSI) and was calculated based on the following two questions [28] (i) ‘How many cigarettes per day do you usually smoke, on the days when you smoke?’ (≤10 (0); 11–20 (1); 21–30 (2); ≥31 (3)) and (ii) ‘How soon after you wake up do you light up your first cigarette, on the days that you smoke?’ (≤5 min (3); 6–30 min (2); 31–60 min (1); >60 min (0)). Responses were dichotomized into light/medium (scores 0–4) vs heavy (scores 5–6) smokers [28]. 

E-cigarette dependence (among vapers at baseline) was assessed through a modified version of the HSI applied to vaping, the Heaviness of Vaping Index (HVI), and included the following two questions: (i) ‘How many times per hour do you use your e-cigarette, on the days that you vape?’ (≤1 (0); 1–5 times (1); 6–10 times (2); nearly all the time (3)) and (ii) ‘Usually, how soon after waking up do you draw your first puff on your e-cigarette, on the days that you vape?’ (≤5 min (3); 6–30 min (2); 31–60 min (1); >60 min (0)). Responses were dichotomized into light/medium (scores 0–4) vs heavy (scores 5–6) vapers.

COVID-19-related characteristics included perceived COVID-19 risk to one’s health, assessed with the question ‘What risk does COVID-19 pose to your health?’. This was dichotomised into major risk or significant risk versus all other (moderate risk, minor risk, no risk at all, don’t know).

We also assessed diagnosed or suspected COVID-19 (measured at 12-months). Participants were asked whether they had been tested for COVID-19 with a swab test (to check current infection) and whether they had been tested for COVID-19 with an antibody/blood test (to check past infection), with the response options (i) yes and tested positive at least once, (ii) yes and tested negative every time, (iii) yes and awaiting results, (iv) no, and (v) prefer not to say for both questions. Participants who reported not having had a positive COVID-19 test were asked “The key symptoms for COVID-19 are high temperature/fever or a new, continuous cough, and loss or change to your sense of smell or taste. Do you think you HAVE or HAD COVID-19?” with the answer options (i) I think I have COVID-19, (ii) I think I had COVID-19, (iii) I do not think I have or have had COVID-19, (iv) don’t know, and (v) prefer not to say. All participants reporting ‘yes and tested positive at least once’ to question 1 and/or 2, or who reported thinking they have or had COVID-19 to question 3 were considered as diagnosed/suspected COVID-19, with all other responses were considered as not diagnosed/suspected COVID-19.

### 2.4. Statistical Analysis

The protocol and analysis plan were pre-registered on Open Science Framework (accessed 17 December 2021. https://osf.io/cdpqf/). Data were analysed in SPSS version 27 (IBM, New York, NY, USA).

Descriptive statistics were calculated to characterise the sample. Independent *t*-tests and chi-squared tests were conducted to assess differences in baseline characteristics between participants identified as smokers/vapers at baseline who completed the 12-month follow up versus those who did not.

For RQ1, we conducted a descriptive analysis of the proportions (and 95% confidence interval (CI)) of smokers/vapers at baseline who reported a quit attempt at any time between the onset of COVID-19 pandemic and the 12-month follow up (i.e., FU1–FU4), and of the proportions (and 95% CI) of participants who were smokers/vapers at baseline and non-smokers/non-vapers at the 12-month follow up.

For RQ2, we calculated the proportions (and 95% CI) of smokers/vapers at baseline who remained smokers/vapers at the 12-month follow up reporting an increase, decrease, and no change in the number of cigarettes smoked per day/frequency of vaping per hour from baseline to the 12-month follow up.

For RQ3, logistic regression analyses were conducted to examine the association of sociodemographic, smoking/vaping, and COVID-19-related characteristics with making a quit attempt versus not (referent) between the onset of the COVID-19 pandemic and the 12-month follow up. Logistic regression analyses were also conducted to examine the association of quitting smoking/vaping successfully versus not (referent) between baseline and a 12-month follow up with potential explanatory covariates (sociodemographic, smoking/vaping, and COVID-19-related characteristics) included in the model. Additionally, multinomial logistic regression analyses were conducted to examine the association of sociodemographic, smoking/vaping, and COVID-19-related characteristics, with (i) decreased smoking/vaping, and (ii) increased smoking/vaping, versus no change (referent).

For RQ4, descriptive analysis of the proportion (and 95% CI) of those citing COVID-19-related reasons for making a quit attempt was conducted. 

## 3. Results

Of the analytic sample (n = 332) of smokers and vapers, 68.7% (228) were smokers and 46.7% (155) were vapers at baseline (among smokers and vapers there were 51 dual users, 15.4%). Overall, participants’ mean age was 49.1 (SD = 13.5), more than half were female (58.1%), the majority were of white ethnicity, most had post-16 education qualifications, and three quarters had an annual income of less than £50,000. Almost half of them reported having a health problem and less than one-third perceived being at high risk of COVID-19, while one third had been diagnosed with/suspected that they had COVID-19 (Table 1). At baseline, most current smokers (93.9%) reported low/medium cigarette dependence compared with 71.6% of vapers reporting low/medium e-cigarette dependence. More than half (58.8%) of smokers and a third (28.4%) of vapers were motivated to quit smoking and vaping, respectively. 

### 3.1. RQ1: Quit Attempts and Quit Rate among Smokers and Vapers

Of the 228 smokers who were followed up successfully at 12 months, 45.0% (95% CI 38.0–51.0%, n = 102) made a quit attempt at any time between the onset of the COVID-19 pandemic and the 12-month follow up, and 17.5% (95% CI 12.6–22.5%, n = 40) had stopped smoking cigarettes at the 12-month follow up.

Of the 155 vapers who were followed-up successfully at 12 months, 25.0% (95% CI 18.0–31.0%, n = 39) made a quit attempt at any time between the onset of the COVID-19 pandemic and the 12-month follow up, and 18.1% (95% CI 11.9–24.2%, n = 28) had quit vaping at the 12-month follow up.

### 3.2. RQ2: Changes in Smoking and Vaping

Among continuing smokers (n = 174), 27.0% (95% CI 20.4–33.7%) increased the number of cigarettes smoked per day, 35.1% (95% CI 27.9–42.2%) decreased, and 37.9% (95% CI 30.7–45.2%) did not change the number of cigarettes smoked per day between baseline and the 12-month follow up.

Among continuing vapers (n = 109), 24.8% (95% CI 16.5–33.0%) increased the frequency of vaping, 27.5% (95% CI 19.0–36.0%) decreased, and 47.7% (95% CI 38.2–55.2%) did not change their vaping frequency between baseline and the 12-month follow up.

### 3.3. RQ3: Factors Associated with Quit Attempts, Quit Rate, and Changes in Smoking and Vaping 

Making a quit attempt between the onset of the COVID-19 pandemic and the 12-month follow up and smoking cessation at the 12-month follow up were both associated with being motivated to quit smoking and being younger (Table 2). No significant predictors were identified for any change observed in the number of cigarettes smoked per day between baseline and the 12-month follow up (Table 3).

No significant predictors were identified for making a vaping quit attempt, while vaping cessation at the 12-month follow up was associated with being a cigarette smoker at baseline (Table 4). Comparisons of exclusive vapers and dual users at baseline showed that more dual users than exclusive vapers quit vaping (29.4% and 12.5% respectively, *p* = 0.01). No significant predictors were identified for any changes observed in the frequency of vaping per hour between baseline and the 12-month follow up (Table 5).

### 3.4. RQ4: Quit Attempts Due to COVID-19-Related Reasons

Of the 103 smokers who made a quit attempt at any time between the onset of the COVID-19 pandemic in the UK and the 12-month follow up, 27.5% (95% CI 13.0–41.9%) reported COVID-19-related reasons for making a quit attempt. The most popular COVID-19 reason was ‘Health problems/concerns related to COVID-19’ (25.0%), followed by ‘Money is tighter due to COVID-19’ (17.5%).

Of the 39 vapers who made a quit attempt at any time between the onset of COVID-19 pandemic in UK and the 12-month follow up, 16.1% (95% CI 10.3–22.0%) reported COVID-19-related reasons for making a quit attempt. Similar to smokers, the most popular COVID-19 reason for vapers was ‘Health problems/concerns related to COVID-19’ (15.4%), followed by ‘Money is tighter due to COVID-19’ (12.8%).

## 4. Discussion

### 4.1. Summary of Findings

Using longitudinal data of a sample of UK smokers and/or vapers adults, we examined quit attempts and quit rate, as well as changes in smoking and vaping during the first year of the COVID-19 pandemic. Results showed that almost half of smokers in our sample made a quit attempt since the onset of the COVID-19 pandemic, while a quarter of vapers tried to quit vaping in the same period. Similar proportions of smokers and vapers (~18%) reported that they quit successfully during the study period. Additionally, similar proportions of smokers reported smoking less or the same number of cigarettes during the study period, while fewer smokers reported an increase in the number of cigarettes smoked per day. Half of the vapers reported no change in the frequency of vaping per hour, while a similar proportion of vapers reported vaping less and more. Motivation to quit smoking was associated with making a quit attempt and quit rate among smokers, while being a cigarette smoker was associated with stopping in vapers. 

### 4.2. Comparison to Previous Research and Implications

Similar to research conducted in the early days of the COVID-19 pandemic (i.e., [11,12,13]), our findings indicate that a substantial number of smokers made a quit attempt during the first year of the COVID-19 pandemic, and 17.5% quit cigarette smoking successfully. It should, however, be noted that such findings are based on a small sample, which is not representative of the UK population. However, data from the Smoking Toolkit Study suggests that in England in 2020, there was an increase in quit attempts compared with 2018, and an increase in the quitting success rate from 14% to 23% [30]. Additionally, Action on Smoking and Health reports that a million people have stopped smoking since the COVID-19 pandemic in Britain [31]. Potential explanations for such changes include that the COVID-19 pandemic and lockdown periods prompted healthy behaviour change, or changes in usual daily routines and social activities providing the opportunity to change smoking behaviour. Indeed, our findings indicate that a substantial proportion of quit attempts were triggered by the COVID-19 pandemic. Cross-sectional studies during the earlier stages of the COVID-19 pandemic also suggest that some cigarette smokers were motivated to quit smoking because of COVID-19 though the proportions were lower than the present findings (e.g., approximately 12% in a representative sample in England; [13]). It can be argued that as the COVID-19 pandemic progressed and disease severity and mortality rates were elevated, people might have been more motivated to follow a healthier lifestyle and tried to quit smoking to protect themselves from COVID-19. Indeed, the most popular COVID-19-related reason for making a quit attempt was health problems or concerns related to COVID-19.

Findings from the present data suggest an association between motivation to quit and quit attempts and quit rate among smokers. Previous research also indicates that motivation to quit is positively associated with quit attempts, while it has been suggested that higher levels of nicotine dependence are negatively associated with quit success in those making an attempt [21,22]. The present sample of smokers had low to medium levels of cigarette addiction, which may be a reason for not finding a significant association between cigarette dependence and quit rate. It was also found that being younger was associated with quit attempts and quit rate. Closures of schools during the COVID-19 lockdowns in the UK meant that children were housebound, and parents might have had fewer opportunities to smoke because of home-schooling and not wanting to expose their kids to second-hand smoking. Additionally, closures of university campuses made many young adults return to their parents’ home, which might have triggered quit attempts and quit success. Indeed, research suggests that many college students paused their smoking and vaping during the first COVID-19 lockdown in the US [32].

Similar proportions of smokers self-reported smoking more or the same numbers of cigarettes per day since the COVID-19 pandemic and fewer smokers self-reported an increase in the numbers of cigarettes smoked per day. Research from the first COVID-19 lockdown in England indicated a higher proportion of smokers increasing their product use [11]. Similarly, research in the US suggests that more smokers increased than decreased their product use during the early stages of COVID-19 pandemic [12,33]. It could be the case that increased stress levels during the beginning of pandemic [34] along with reports that nicotine may be protective against COVID-19 [35], may have triggered increases in smoking in the early days of the pandemic. A clearer understanding of the impact of smoking on COVID-19 outcomes and reductions in stress levels during the later stages of the pandemic might have motivated smokers to reduce cigarette smoking.

Among vapers, our findings indicate that a quarter made a quit attempt and most of them (72%) reported quit success. However, quit rate was associated with being a cigarette smoker, and comparison of exclusive vapers with dual users indicated that more dual users quit vaping since COVID-19, possibly because they simply switched back to smoking. In continuing vapers, we found that this longer-term 12-month study confirmed findings from an earlier short-term study at the beginning of the pandemic [10], showing that the majority did not change their vaping use. However, a higher proportion increased vaping in the early stages of the pandemic, probably because they were staying at home where there are fewer or no restrictions, with more opportunities to vape, and only 10% decreased their vaping during the first lockdown in UK compared with around a quarter a year later. Our results also suggest that a minority of vapers (and smaller proportion than smokers) were motivated to change their product use due to the COVID-19 pandemic, reflecting previous work [10,36]. Such findings may be attributed to the contradictory media reports on the possibility of nicotine being protective against COVID-19 [35] and reports that nicotine-containing vaping is generally safer than cigarette smoking [37], as well as inconclusive findings regarding the association of vaping with COVID-19 infection, disease severity, and death [9,10]. 

### 4.3. Strengths and Limitations

The present study has several strengths. It is one of only a few reporting changes in smoking and vaping over a 12-month period during the pandemic in the UK. Much of the available work assessing the effect of the COVID-19 pandemic on smoking and vaping is dependent on cross-sectional, retrospective studies, which have the potential for recall bias. Furthermore, the variety of measures collected is another advantage of the present study, permitting a detailed analysis of a broad range of potential correlates of changes in smoking and vaping during the COVID-19 pandemic. The measure of quit attempts over multiple points during the 12-month follow up is another strength of the study, as a large proportion of unsuccessful quit attempts fail to be reported if they last a short time and occurred long ago [38]. However, the study also had several limitations. First, smoking and vaping status and product use were exclusively self-reported, though self-reporting of smoking behaviours in low-demand surveys has been shown to be reliable [39]. Second, there is the possibility of selection bias, as the sample was self-selected, and there were also differences between the baseline and follow-up samples. Additionally, people who participated in the study examining the influence of COVID-19 on health behaviours may have had a greater interest in helping to tackle the pandemic than the general population. Third, for some analyses the sample size was small, resulting in wide confidence intervals. Future research is needed to examine changes in cigarette smoking and vaping in representative samples and to examine how COVID-19 may stimulate interest in reducing or quitting smoking and vaping as well as initiation of these behaviours and serve as a novel opportunity to promote cessation or harm reduction during the current and future respiratory viral pandemics.

## 5. Conclusions

In conclusion, our findings suggest that many smokers and vapers have attempted to stop either smoking or vaping, though the number of smokers was greater than vapers, and a high proportion were successful. Additionally, most smokers reported a decrease or no change in cigarette consumption during the COVID-19 pandemic. Similarly, half of vapers reported no change in their vaping consumption and a quarter of them reported vaping less. On the one hand, the pandemic has provided motivation to stop smoking in particular, on the other it may have pushed vapers to switch back to smoking.

## Figures and Tables

**Table 1 ijerph-19-00630-t001:** Baseline sample characteristics of total, followed-up at 12-months, and lost to follow-up samples.

	Total Sample N = 751	Followed-up at 12-Months N = 332	Lost to Follow-up N = 419	*p*
N		332		
Age in years, M (SD)	44.7 (15.3)	49.1 (13.5)	41.3 (15.7)	**<0.001**
Female sex, % (n)	56.1 (420)	58.1 (193)	54.4 (227)	0.311
White ethnicity, % (n)	93.9 (704)	96.7 (321)	91.6 (383)	**0.004**
No post-16 education qualifications, % (n)	22.4 (168)	17.5 (58)	26.3 (110)	**0.004**
Income, % (n)				
≥£50,000, % (n)	24.8 (186)	27.8 (92)	22.4 (94)	0.209
<£50,000, % (n)	65.7 (493)	63.7 (211)	67.3 (282)	
Prefer not to say, % (n)	9.5 (72)	8.5 (29)	10.3 (43)	
Health problems, % (n)	46.6 (341)	47.0 (155)	46.4 (186)	0.874
Perceived high risk of COVID-19, % (n)	25.4 (186)	27.9 (92)	23.4 (94)	0.164
Diagnosed/suspected COVID-19, % (n)	29.3 (215)	30.4 (101)	28.4 (114)	0.541
Current smokers, % (n)	74.0 (556)	68.7 (228)	78.3 (328)	**0.003**
Current vapers, % (n)	44.9 (337)	46.7 (155)	43.4 (182)	0.374

M = mean, SD = standard deviation. Bold indicates statistical significance.

**Table 2 ijerph-19-00630-t002:** Predictors of making a quit attempt and smoking cessation at 12-month follow up among smokers at baseline (N = 228).

	Quit Attempt					Smoking Cessation				
	% (n) [95% CI]	OR [95% CI]	*p*	aOR [95% CI]	*p*	% (n) [95% CI]	OR [95% CI]	*p*	aOR [95% CI]	*p*
Sex: other	45.2 (103) [37.9–53.1]	1 (ref)		1 (ref)		17.3 (39) [8.9–25.7]	1 (ref)		1 (ref)	
Female	44.8 (102) [33.1–56.4]	1.02 [0.59–1.76]	0.947	0.90 [0.47–1.72]	0.756	17.7 (40) [11.5–23.9]	1.03 [0.50–2.10]	0.939	1.17 [0.46–2.96]	0.743
Ethnicity: other	40.1 (91) [3.9–77.1]	1 (ref)		1 (ref)		20.2 (46) [−10.2–50.2]	1 (ref)		1 (ref)	
White	45.2 (103) [38.2–52.1]	1.23 [0.34–4.46]	0.758	1.04 [0.17–6.29]	0.970	17.3 (39) [12.4–22.5]	0.84 [0.17–4.14]	0.835	0.37 [0.04–3.33]	0.374
Post 16 qualifications: yes	47.5 (108) [39.8–54.5]	1 (ref)		1 (ref)		19.4 (44) [13.7–25.0]	1 (ref)		1 (ref)	
No	35.8 (81) [19.9–51.2]	0.62 [0.30–1.29]	0.202	0.71 [0.30–1.68]	0.44	8.1 (18) [−1.1–17.3]	0.37 [0.11–1.26]	0.112	0.71 [ 0.18–2.89]	0.636
Income: ≥£50,000	40.2 (92) [27.1–53.2]	1 (ref)		1 (ref)		17.3 (39) [7.2–27.3]	1 (ref)		1 (ref)	
<£50,000	47.1 (107) [39.9–55.5]	1.35 [0.73–2.50]	0.346	1.67 [0.80–3.46]	0.170	17.3 (39) [10.9–23.2]	0.98 [0.44–2.20]	0.968	1.24 [0.47–3.29]	0.661
Prefer not to say	45.6 (104) [23.8–68.4]	1.27 [0.47–3.41]	0.638	1.04 [0.33–3.31]	0.950	22.7 (52) [3.7–41.8]	1.41 [0.42–4.72]	0.576	1.21 [0.28–5.25]	0.803
Health problems: yes	43.8 (103) [34.9–53.3]	1 (ref)		1 (ref)		12.3 (28) [5.9–18.6]	1 (ref)		1 (ref)	
No	47.2 (99) [38.7–56.2]	1.14 [0.68–1.93]	0.622	1.21 [0.61–2.39]	0.970	22.5 (51) [14.9–30.1]	2.08 [1.01–4.27]	**0.047**	2.21 [0.78–6.23]	0.134
Perceived high risk of COVID-19: no	45.7 (104) [37.9–53.4]	1 (ref)		1 (ref)		20.2 (46) [13.7–26.3]	1 (ref)		1 (ref)	
Yes	45.2 (103) [33.4–58.0]	1.02 [0.57–1.81]	0.950	1.53 [0.74–3.15]	0.251	12.1 (28) [20.2–7.9]	0.55 [0.24–1.27]	0.163	1.03 [0.34–3.15]	0.955
Confirmed/suspected COVID-19: no	43.4 (99) [35.8–50.2]	1 (ref)		1 (ref)		17.5 (40) [11.6–23.4]	1 (ref)		1 (ref)	
Yes	50.1 (114) [38.9–62.0]	1.35 [0.77–2.39]	0.298	1.28 [0.64–2.54]	0.489	17.7 (40) [8.4–26.9]	1.01 [0.48–2.13]	0.979	0.44 [0.16–1.23]	0.118
Vaping status: non-vaper	41.3 (94) [34.6–49.2]	1 (ref)		1 (ref)		15.8 (36) [10.4–21.3]	1 (ref)		1 (ref)	
Vaper	57.1 (130) [43.7–71.3]	1.88 [1.00–3.53]	0.050	1.45 [0.71–2.97]	0.313	23.5 (54) [11.5–35.6]	1.64 [0.76–3.51]	0.205	0.80 [0.29–2.20]	0.669
Light/medium smoker	45.2 (103) [38.6–52.1]	1 (ref)		1 (ref)		16.8 (38) [10.9–21.2]	1 (ref)		1 (ref)	
Heavy smoker	46.1 (105) [15.6–78.1]	1.01 [0.34–3.27]	0.920	1.11 [0.28–4.45]	0.882	7.7 (17) [−9.1–24.5]	0.44 [0.06–3.46]	0.432	1.23 [0.12–12.97]	0.863
Motivation to quit: low	29.2 (67) [18.1–39.4]	1 (ref)		1 (ref)		7.1 (16) [1.0–13.3]	1 (ref)		1 (ref)	
High	52.0 (119) [44.1–61.3]	2.73 [1.47–5.08]	**0.001**	2.65 [1.39–5.06]	**0.003**	20.2 (46) [13.3–27.0]	3.28 [1.20–8.94]	**0.020**	4.36 [1.46–13.02]	**0.008**
	**M (SD)**	**OR** **[95% CI]**	** *p* **	**aOR** **[95% CI]**	** *p* **	**%** **[95% CI]**	**OR** **[95% CI]**	** *p* **	**aOR** **[95% CI]**	** *p* **
Age (cont.)	45.9 (15.2)	0.76 [0.57–1.01]	0.055	0.65 [0.45–0.93]	**0.018**	40.9 (14.7)	0.54 [0.38–0.78]	**0.001**	0.45 [0.27–0.77]	**0.003**

CI, confidence interval. OR, odds ratio. aOR, adjusted odds ratio. M, mean. SD, standard deviation. Age was standardised. cont., continuous. Multivariable model: age, sex, ethnicity, post-16 qualifications, income, vaping status, health problems, perceived risk of COVID-19, confirmed/suspected COVID-19, nicotine dependence, motivation to quit. Missing data is due to attrition from the survey. Bold indicates statistical significance.

**Table 3 ijerph-19-00630-t003:** Correlates of smoking less and smoking more numbers of cigarettes per day between baseline and the 12-month follow-up among current smokers at both time points (N = 174).

	Smoking Less					Smoking More				
	% [95% CI]	OR [95% CI]	*p*	aOR [95% CI]	*p*	% [95% CI]	OR [95% CI]	*p*	aOR [95% CI]	*p*
Sex: other	40.4 [27.2–53.5]	1 (ref)		1 (ref)		17.5 [7.4–27.7]	1 (ref)		1 (ref)	
Female	32.5 [23.9–41.1]	1.06 [0.52–2.18]	0.876	1.21 [0.54–2.74]	0.643	31.6 [23.1–40.2]	0.47 [0.20–1.12]	0.088	0.41 [0.16–1.09]	0.073
Ethnicity: other	40.0 [−28.0–108.0]	1 (ref)		1 (ref)		60.0 [−8.0–128]	1 (ref)		1 (ref)	
White	34.9 [27.7–42.2]	0	1	0	1	26.0 [19.4–32.7]	0	1	0	1
Post 16 qualifications: yes	33.3 [25.5–41.2]	1 (ref)		1 (ref)		27.7 [20.2–35.1]	1 (ref)		1 (ref)	
No	42.4 [24.6–60.2]	0.67 [0.28–1.62]	0.375	0.50 [0.18–1.38]	0.182	24.2 [8.8–96.7]	0.97 [0.36–2.65]	0.960	0.55 [0.17–1.81]	0.324
Income: ≥£50,000	34.1 [19.5–48.7]	1 (ref)		1 (ref)		25.0 [11.7–38.3]	1 (ref)		1 (ref)	
<£50,000	34.5 [25.7–43.3]	0.42 [0.09–1.96]	0.267	0.57 [0.11–3.06]	0.515	26.7 [18.6–34.9]	0.37 [0.07–1.85]	0.224	0.37 [0.06–2.21]	0.278
Prefer not to say	42.9 [13.2–72.5]	0.44 [0.10–1.89]	0.27	0.53 [0.11–2.49]	0.418	35.7 [7.0–64.4]	0.41 [0.09–1.86]	0.249	0.62 [0.12–2.21]	0.566
Health problems: yes	40.0 29.7–50.3]	1 (ref)		1 (ref)		23.3 [14.4–32.2]	1 (ref)		1 (ref)	
No	29.8 [19.8–39.7]	0.69 [0.34–1.40]	0.309	0.82 [0.36–1.85]	0.621	31.0 [20.9–41.1]	1.24 [0.58–2.62]	0.577	1.18 [0.48–2.93]	0.708
Perceived high risk of COVID-19: no	30.3 [21.9–38.6]	1 (ref)		1 (ref)		31.9 [23.4–40.4]	1 (ref)		1 (ref)	
Yes	45.5 [31.9–59.0]	1.49 [0.72–3.08]	0.284	1.05 [0.44–2.50]	0.908	16.4 [6.3–26.5]	0.51 [0.21–1.24]	0.136	0.41 [0.14–1.19]	0.102
Confirmed/suspected COVID-19: no	33.9 [25.4–42.3]	1 (ref)		1 (ref)		26.6 [18.7–34.5]	1 (ref)		1 (ref)	
Yes	38.0 [24.1–51.9]	1.30 [0.60–2.82]	0.501	1.19 [0.49–2.91]	0.688	28.0 [15.1–40.9]	1.22 [0.53–2.82]	0.636	1.07 [0.41–2.82]	0.89
Vaping status: non-vaper	37.7 [29.5–45.9]	1 (ref)		1 (ref)		23.9 [16.7–31.1]	1 (ref)		1 (ref)	
Vaper	25.0 [10.1–39.9]	0.71 [0.28–1.79]	0.463	0.82 [0.31–2.17	0.685	38.9 [22.2–55.6]	1.73 [0.72–4.13]	0.218	1.69 [0.65–4.42]	0.286
Light/medium smoker	32.7 [25.4–40.0]	1 (ref)		1 (ref)		28.4 [21.4–35.4]	1 (ref)		1 (ref)	
Heavy smoker	66.7 [35.4–97.9]	0.315 [0.08–1.25]	0.100	0.33 [0.06–1.96]	0.223	8.3 [−10.0–26.7]	2.19 [0.22–21.74]	0.503	1.71 [0.09–33.69]	0.724
Motivation to quit: low	28.7 [26.2–51.2]	1 (ref)		1 (ref)		21.0 [10.6–31.4]	1 (ref)		1 (ref)	
High	33.3 [24.2–42.5]	1.02 [0.49–2.10]	0.968	1.07 [0.50–2.30]	0.855	31.4 [22.4–40.5]	0.58 [0.26–1.32]	0.196	0.53 [0.22–1.28]	0.530
	**M (SD)**	**OR** **[95% CI]**	** *p* **	**aOR** **[95% CI]**	** *p* **	**%** **[95% CI]**	**OR** **[95% CI]**	** *p* **	**aOR** **[95% CI]**	** *p* **
Age (cont.)	52.2 (13.1)	1.02 [0.67–1.55]	0.920	1.09 [0.66–1.79]	0.733	45.4 (13.9)	0.59 [0.38–0.90]	**0.014**	0.72 [0.43–1.19]	0.200

Reference category: no change. CI, confidence interval. OR, odds ratio. aOR, adjusted odds ratio. M, mean. SD, standard deviation. Age was standardised. cont., continuous. Multivariable model: age, sex, ethnicity, post-16 qualifications, income, vaping status, health problems, perceived risk of COVID-19, confirmed/suspected COVID-19, nicotine dependence, motivation to quit. Missing data is due to attrition from the survey. Bold indicates statistical significance.

**Table 4 ijerph-19-00630-t004:** Predictors of making a quit attempt and vaping cessation at 12-month follow up among vapers at baseline (N = 155).

	Quit Attempt					Vaping Cessation				
	% (n) [95% CI]	OR [95% CI]	*p*	aOR [95% CI]	*p*	% (n) [95% CI]	OR [95% CI]	*p*	aOR [95% CI]	*p*
Sex: other	24.5 (38) [15.3–34.1]	1 (ref)		1 (ref)		20.5 (32) [11.4–29.7]	1 (ref)		1 (ref)	
Female	25.6 (40) [15.3–35.2]	1.02 [0.49–2.12]	0.963	0.90 [0.39–2.10]	0.810	15.6 (24) [7.3–23.9]	0.72 [0.31–1.63]	0.426	0.59 [0.22–1.58]	0.295
Ethnicity: other	0	1 (ref)		1 (ref)		0	1 (ref)		1 (ref)	
White	24.2 (38) [17.2–31.3]	0	1	0	1	17.5 (27) [11.5–23.6]	0	1	0	1
Post 16 qualifications: yes	25.3 (39) [17.0–33.5]	1 (ref)		1 (ref)		17.9 (28) [11.2–24.7]	1 (ref)		1 (ref)	
No	22.5 (35) [5.2–39.1]	0.86 [0.32–2.31]	0.761	1.04 [0.33–3.30]	0.954	18.5 (29) [2.9–34.2]	1.04 [0.36–3.03]	0.946	1.19 [0.33–4.31]	0.793
Income: ≥£50,000	24.8 (38) [11.3–38.2]	1 (ref)		1 (ref)		17.9 (28) [6.2–29.4]	1 (ref)		1 (ref)	
<£50,000	24.3 (37) [16.5–33.2]	0.99 [0.44–2.25]	0.979	1.29 [0.51–3.26]	0.596	17.5 (27) [9.6–24.7]	0.96 [0.38–2.42]	0.929	1.11 [0.39–3.17]	0.852
Prefer not to say	27.1 (42) [−4.2–29.1]	1.16 [0.26–5.15]	0.846	2.10 [0.34–12.98]	0.423	27.3 (42) [−4.1–58.6]	1.73 [0.38–8.02]	0.481	2.89 [0.45–18.45]	0.261
Health problems: yes	26.1 (40) [15.5–36.2]	1 (ref)		1 (ref)		16.2 (25) [7.6–24.8]	1 (ref)		1 (ref)	
No	23.2 (36) [14.8–33.2]	0.89 [0.43–1.85]	0.748	0.93 [0.38–2.25]	0.870	19.8 (31) [10.9–28.6]	1.27 [0.56–2.90]	0.568	1.29 [0.48–3.54]	0.613
Perceived high risk of COVID-19: no	26.8 (42) [18.3–34.2]	1 (ref)		1 (ref)		19.8 (31) [12.4–26.9]	1 (ref)		1 (ref)	
Yes	21.2 (33) [7.9–35.3]	0.77 [0.32–1.87]	0.568	1.03 [0.37–2.88]	0.959	13.2 (20) [1.9–24.4]	0.62 [0.22–1.76]	0.369	0.84 [0.25–2.84]	0.780
Confirmed/suspected COVID-19: no	22.8 (35) [14.3–30.8]	1 (ref)		1 (ref)		17.5 (27) [10.1–24.5]	1 (ref)		1 (ref)	
Yes	31.1 (48) [17.2–45.1]	1.62 [0.74–3.52]	0.224	1.36 [0.56–3.32]	0.497	20.0 (31) [7.9–32.2]	1.20 [0.50–2.89]	0.689	0.99 [0.35–2.84]	0.988
Smoking status: non-smoker	20.8 (32) [12.5–28.0]	1 (ref)		1 (ref)		12.5 (19) [6.0–19.0]	1 (ref)		1 (ref)	
Smoker	33.2 (51) [20.7–47.5]	1.98 [0.93–4.20]	0.077	1.70 [0.70–4.08]	0.239	29.4 (46) [16.5–42.4]	2.92 [1.26–6.74]	**0.012**	3.09 [1.15–8.30]	**0.025**
Light/medium vaper	29.5 (40) [20.9–37.2]	1 (ref)		1 (ref)		19.8 (31) [12.3–27.4]	1 (ref)		1 (ref)	
Heavy vaper	18.3 (28) [3.9–33.1]	0.54 [0.19–1.54]	0.246	0.69 [0.21–2.30]	0.543	17.9 (28) [2.7–32.9]	0.88 [0.30–2.57]	0.815	1.16 [0.33–4.11]	0.823
Motivation to quit: low	20.1 (31) [12.8–28.1]	1 (ref)		1 (ref)		15.8 (24) [8.6–23.1]	1 (ref)		1 (ref)	
High	39.8 (62) [24.2–54.1]	2.55 [1.17–5.56]	**0.019**	2.23 [0.95–5.24]	0.066	25.0 (39) [11.7–38.3]	1.77 [0.74–4.21]	0.196	1.66 [0.61–4.47]	0.319
	**M (SD)**	**OR** **[95% CI]**	** *p* **	**aOR** **[95% CI]**	** *p* **	**%** **[95% CI]**	**OR** **[95% CI]**	** *p* **	**aOR** **[95% CI]**	** *p* **
Age (cont.)	45.9 (13.5)	0.59 [0.38–0.92]	**0.020**	0.74 [0.44–1.25]	0.261	45.7 (15.0)	0.61 [0.37–0.98]	0.041	0.85 [0.48–1.51]	0.582

CI, confidence interval. OR, odds ratio. aOR, adjusted odds ratio. M, mean. SD, standard deviation. Age was standardised. cont., continuous. Multivariable model: age, sex, ethnicity, post-16 qualifications, income, smoking status, health problems, perceived risk of COVID-19, confirmed/suspected COVID-19, nicotine dependence, motivation to quit. Missing data is due to attrition from the survey. Bold indicates statistical significance.

**Table 5 ijerph-19-00630-t005:** Correlates of vaping less and vaping more frequently per hour per day between baseline and 12-month follow-up among current vapers at both time points (N = 109).

		Vaping Less					Vaping More			
	% [95% CI]	OR [95% CI]	*p*	OR_adj_ [95% CI]	*p*	% [95% CI]	OR [95% CI]	*p*	OR_adj_ [95% CI]	*p*
Sex: other	24.1 [12.3–35.9]	1 (ref)		1 (ref)		22.2 [10.8–33.7]	1 (ref)		1 (ref)	
Female	30.9 [18.3–43.5]	0.60 [0.25–1.50]	0.279	0.82 [0.27–2.48]	0.729	24.3 [15.1–39.4]	0.63 [0.25–1.62]	0.341	0.67 [0.24–1.89]	0.453
Ethnicity: other	0	1 (ref)		1 (ref)		0	1 (ref)		1 (ref)	
White	27.5 [19.0–36.0]	0	1	0	1	24.8 [16.5–33.0]	0	1	0	1
Post 16 qualifications: yes	27.8 [18.3–37.2]	1 (ref)		1 (ref)		25.6 [16.4–34.7]	1 (ref)		1 (ref)	
No	26.3 [4.5–48.1]	1.19 [0.36–3.88]	0.773	1.33 [0.32–5.49]	0.696	21.1 [0.9–41.2]	1.37 [0.39–4.85]	0.627	1.02 [0.25–4.07]	0.980
Income: ≥£50,000	24.2 [8.8–39.7]	1 (ref)		1 (ref)		27.3 [11.2–43.3]	1 (ref)		1 (ref)	
<£50,000	29.6 [18.7–40.5]	1.50 [0.13–16.82]	0.742	0	1	23.9 [13.8–34.1]	1.69 [0.15–18.71]	0.670	2.21 [0.16–34.04]	0.541
Prefer not to say	20.0 [−35.6–75.5]	1.91 [0.19–19.59]	0.586	0	1	20.0 [−35.5–75.6]	1.54 [0.15–16.01]	0.715	2.57 [0.17–38.19]	0.493
Health problems: yes	27.3 [15.1–39.4]	1 (ref)		1 (ref)		23.6 [12.1–35.2]	1 (ref)		1 (ref)	
No	27.8 [15.4–40.1]	0.92 [0.38–2.27]	0.867	0.78 [0.26–2.40]	0.670	25.9 [13.9–38.0]	0.86 [0.34–2.18]	0.750	0.91 [0.62–2.66]	0.866
Perceived high risk of COVID-19: no	24.1 [14.7–33.5]	1 (ref)		1 (ref)		27.7 [17.9–37.5]	1 (ref)		1 (ref)	
Yes	38.5 [18.4–58.5]	1.67 [0.61–4.55]	0.315	2.56 [0.75–9.09]	0.133	15.4 [0.5–30.3]	0.58 [0.17–2.00]	0.390	0.75 [0.18–3.07]	0.692
Confirmed/suspected COVID-19: no	33.3 [22.6–44.0]	1 (ref)		1 (ref)		21.8 [12.4–31.2]	1 (ref)		1 (ref)	
Yes	12.9 [0.4–25.4]	3.16 [0.95–10.50]	0.061	3.84 [1.00–14.65]	0.059	32.3 [14.8–49.7]	0.83 [0.31–2.18]	0.700	0.86 [0.29–2.54]	0.790
Smoking status: non-smoker	25.0 [15.6–34.5]	1 (ref)		1 (ref)		22.6 [13.5–31.8]	1 (ref)		1 (ref)	
Smoker	36.0 [15.8–56.2]	2.35 [0.79–6.99]	0.121	2.04 [0.57–7.30]	0.275	32.0 [12.4–51.7]	2.31 [0.76–7.09]	0.141	1.44 [0.42–4.93]	0.558
Light/medium vaper	24.7 [15.1–34.3]	1 (ref)		1 (ref)		29.6 [19.5–39.8]	1 (ref)		1 (ref)	
Heavy vaper	27.3 [7.1–47.5]	0.85 [0.28–2.59]	0.780	0.62 [0.15–2.50]	0.502	13.6 [−1.9–29.2]	0.36 [0.09–1.38]	0.135	0.38 [0.08–1.82]	0.226
Motivation to quit: low	25.3 [15.5–35.1]	1 (ref)		1 (ref)		25.3 [15.5–35.1]	1 (ref)		1 (ref)	
High	33.3 [15.4–51.2]	1.50 [0.56–4.02]	0.420	1.05 [0.31–3.64]	0.927	23.3 [7.3–39.4]	1.05 [0.36–3.05]	0.928	0.70 [0.21–2.29]	0.544
	**M (SD)**	**OR** **[95% CI]**	** *p* **	**aOR** **[95% CI]**	** *p* **	**%** **[95% CI]**	**OR** **[95% CI]**	** *p* **	**aOR** **[95% CI]**	** *p* **
Age (cont.)	50.2 (12.0)	0.67 [0.35–1.27]	0.221	0.48 [0.21–1.08]	0.077	48.4 (11.5)	0.55 [0.28–1.05]	0.069	0.52 [0.24–1.11]	0.091

Reference category: no change. CI, confidence interval. OR, odds ratio. aOR, adjusted odds ratio. M, mean. SD, standard deviation. Age was standardised. cont., continuous. Multivariable model: age, sex, ethnicity, post-16 qualifications, income, smoking status, health problems, perceived risk of COVID-19, confirmed/suspected COVID-19, nicotine dependence, motivation to quit. Missing data is due to attrition from the survey. Bold indicates statistical significance.

## Data Availability

Data is available upon request.

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
