# Peer review of "Changes in Cigarette Smoking and Vaping in Response to the COVID-19 Pandemic in the UK: Findings from Baseline and 12-Month Follow up of HEBECO Study"

_ijerph, 2022, doi:10.3390/ijerph19020630_

Round 1

Reviewer 1 Report

Congratulations for the article. Very interesting and provides evidence for policy-making.

In the methods section, you mention a logistic analysis, considering "attempts to quit" as the dependent variable?, Please provide informtion about how many persons attempt to quit smoking&vaping during follow-up. In results, please include a descriptive table with this information, desagregated by the different variables you included in the regression models.

In the results you exhibit a lost superior to 50%, especially smokers:

  1. How did you define your "cohort"? By the exposure to COVID-19?. Please include it in the text
  2. How did you follow-up you "cohort"?. Please include it in the text.
  3. Did you have the chance to ask the reasons for not continuing in the study? Please include it in the text.
  4. Please discuss how this can affect you estimations (under / over), considering the wide 95% CI you show in your tables. Please include it in the text.

In the discussion section, you provide qualitative analysis of "self-reported" smoking-vaping, nevertheless you do not give antecedents to understand if in your study, the prevalences/risks are under or over estimated. Please provide.

Author Response

Dear Reviewer,

Thank you for taking the time to read and comment on our manuscript. I appreciated your constructive criticism and advice. We have amended the paper in light of your suggestions as detailed in italics below.

Congratulations for the article. Very interesting and provides evidence for policy-making.

Thank you for your positive comments.

In the methods section, you mention a logistic analysis, considering "attempts to quit" as the dependent variable?, Please provide information about how many persons attempt to quit smoking&vaping during follow-up. In results, please include a descriptive table with this information, desagregated by the different variables you included in the regression models.

We have now included the number of participants who made a quit attempt and quit successfully both in the results section ‘Of the 228 smokers who were followed-up successfully at 12-months, 45.0% (95%CI 38.0-51.0%, n=102) made a quit attempt at any time between the onset of the COVID-19 pandemic and the 12-month follow-up, and 17.5% (95%CI 12.6-22.5%, n=40) had stopped smoking cigarettes at 12-month follow-up.

Of the 155 vapers who were followed-up successfully at 12 months, 25.0% (95%CI 18.0-31.0%, n=39) made a quit attempt at any time between the onset of COVID-19 pandemic and the 12-month follow-up, and 18.1% (95%CI 11.9-24.2%, n=28) had quit vaping at 12-months follow-up.’ (Lines 292,293,296, 297, page 6) and in the Tables 2 and 4 as suggested.

In the results you exhibit a lost superior to 50%, especially smokers:

Of the eligible 751 participants at baseline, 332 (44.2%) were successfully followed-up at 12 months. We also compared those who were followed-up with those lost to follow-up. ‘Participants who were successfully followed-up at 12 months were significantly older (p<0.001), more were of white ethnicity (p=0.004), with post-16 education qualifications (p=0.004), and non-smokers (p=0.003) than those who did not complete the 12-month follow-up’. Participants were followed up via email, with up to three reminders to complete the survey sent at each follow-up, though there was no incentive to complete the follow-up surveys in order to boost retention rate.

How did you define your "cohort"? By the exposure to COVID-19?. Please include it in the text.

We included all participants who responded our survey, irrespective of exposure status. They did have to be baseline vapers or smokers. In section 2.2, page 3, lines 115-123 we describe the study sample ‘This study included a self-selected sample of UK–based adult (18+ years), who were either smokers or vapers (some were both smokers and vapers; dual users) and completed the baseline survey of the HEBECO study between 23rd April 2020 (initiation of recruitment) and June 14th 2020 inclusive (marking the end of the first national UK lockdown) and were successfully followed-up after 12-months (FU4; ease of COVID-19-related re-strictions in UK). A total of 2994 participants completed the baseline questionnaire of the HEBECO study, of whom 751 were smokers and vapers (556 smokers; 337 vapers; among the smokers and vapers there were 142 dual users), who were potentially eligible for this study. Of these eligible 751 participants at baseline, 332 (44.2%) were successfully followed-up at 12 months’.

We also mention the recruitment strategy of HEBECO study, ‘Initial recruitment at baseline was online and involved sharing study invitations via multiple channels, including unpaid and paid advertisements on social media (e.g., Facebook, Twitter, Reddit), an email campaign across the network of UCL, other universities in UK, Public Health England, Cancer Research UK, charities, and local authorities across the UK. The full recruitment strategy is available online (https://osf.io/sbgru/).’ lines 127-131, page 3.

We hope that these details make our cohort sufficiently clear.

How did you follow-up you "cohort"?. Please include it in the text.

In section 2.2, page 3, lines 133-135, we describe how participants were followed-up ‘Participants were followed up via email (except for participants who explicitly opted out), with up to three reminders to complete the survey sent at each follow-up.’

Did you have the chance to ask the reasons for not continuing in the study? Please include it in the text.

Participants were followed up via email to complete the follow-up surveys. We were unable to ask participants the reasons for not completing the follow-up surveys (as this may be considered unethical). We added the following sentence to the manuscript ‘Reasons for not completing the follow-up surveys were not assessed.’ Lines 135-136, page 3.

Please discuss how this can affect you estimations (under / over), considering the wide 95% CI you show in your tables. Please include it in the text.

In the limitations of the study, we mention that ‘the sample size was small, resulting in wide confidence intervals’. line 458, page 14.

In the discussion section, you provide qualitative analysis of "self-reported" smoking-vaping, nevertheless you do not give antecedents to understand if in your study, the prevalences/risks are under or over estimated. Please provide.

In the limitations of the study, we mention that ‘smoking and vaping status and product use were exclusively self-reported, though self-reporting of smoking behaviours in low-demand surveys has been shown to be reliable [39].’ Lines 451-453, page 14. 

Reviewer 2 Report

I have read the article entitled "Changes in cigarette smoking and vaping in response to the COVID-19 pandemic in UK: Findings from baseline and 12-month follow-up of HEBECO study". This is a very interesting study that meets the editorial requirements. The entire manuscript is written in a clear and accessible way. The authors described in detail all aspects of the study and the results obtained. In light of such extensive previous sections of the manuscript, the Conclusions section needs to be expanded as its volume is disproportionate to the rest of the article.

Author Response

Dear Reviewer,

Thank you for taking the time to read and comment on our manuscript. I appreciated your constructive criticism and advice. We have amended the paper in light of your suggestions as detailed in italics below.

I have read the article entitled "Changes in cigarette smoking and vaping in response to the COVID-19 pandemic in UK: Findings from baseline and 12-month follow-up of HEBECO study". This is a very interesting study that meets the editorial requirements. The entire manuscript is written in a clear and accessible way. The authors described in detail all aspects of the study and the results obtained. In light of such extensive previous sections of the manuscript, the Conclusions section needs to be expanded as its volume is disproportionate to the rest of the article.

Thank you for your comments. We added the following to the conclusion section ‘Additionally, most smokers reported a decrease or no change in cigarette consumption during the COVID-19 pandemic. Similarly, half of vapers reported no change in their vaping consumption and a quarter of them reported vaping less’. Lines 466-469, page 15.

Reviewer 3 Report

Abstract: Where it says "due to COVID-19" please clarify whether this means they quit because they were diagnosed/contracted COVID-19?

If there were 5 surveys, which were used for the results presented in the abstract? Is it if they reported a quit attempt at any of the 5 survey time points?

Intro: "This study uses data collected mainly at two timepoints over 12-months to investi-87 gate the long-term effects of the COVID-19 pandemic on smoking and vaping in UK adults." The abstract says there are 5 time points.

Methods: It is unclear why the description of the sample in section 2.2 and the first Table includes those who were not smokers or vapers at baseline since this is indicated as the eligibility criteria for the study.  Did you follow nonsmokers at baseline and see if anyone started smoking/vaping?

Author Response

Dear Reviewer,

Thank you for taking the time to read and comment on our manuscript. I appreciated your constructive criticism and advice. We have amended the paper in light of your suggestions as detailed in italics below.

Abstract: Where it says "due to COVID-19" please clarify whether this means they quit because they were diagnosed/contracted COVID-19?

Participants reported COVID-19 related reason for a quit attempt and not if they quit because they were diagnosed/contracted COVID-19. We have amended the abstract to reflect this. ‘due to COVID-19-related reasons’ lines 17 and 20, page 1.

In the Materials and Methods section, we also describe the COVID-19 related reasons ‘Reasons for making a smoking/vaping quit attempt (assessed among smokers and vapers at baseline who made a quit attempt) assessed at all waves and included: (1) Rules around social distancing/self-isolation due to COVID-19, (2) Children/parents moved back home due to COVID-19, (3) Money is tighter due to COVID-19, (4) Decided it was too expensive, (5) Health problems/concerns related to COVID-19, (6) Health problems/concerns unrelated to COVID-19, (7) Advice from a GP, (8) Government/Tv/radio/press advert, (9) Social campaign, (10) Being contacted by local NHS Stop Smoking Services, (11) Being faced with restrictions already before COVID-19, (12) I knew someone else who was stopping, (13) Seeing a health warning on a packet, (14) Something said by family/friends/children, (15) Improve fitness, (16) Other. Participants could select one or more reasons. Participants who selected at least one of reasons 1, 2, 3, or 5 were classified as motivated to quit smoking/vaping due to COVID-19-related reasons.’ Lines 187-198, page 4.

If there were 5 surveys, which were used for the results presented in the abstract? Is it if they reported a quit attempt at any of the 5 survey time points?

In the present study we used data from baseline and 12-month follow-up surveys, though data on quit attempts were taken from all 5 surveys. The measure of quit attempts over multiple points during the 12-month follow-up is a strength of the study, as a large proportion of unsuccessful quit attempts fail to be reported if they last a short time and occurred longer ago. More information regarding data used is documented in the Materials and Methods section. 2.3 Measures. Lines 136-197, pages 3-4.

Intro: "This study uses data collected mainly at two timepoints over 12-months to investigate the long-term effects of the COVID-19 pandemic on smoking and vaping in UK adults." The abstract says there are 5 time points.

The outcome measures of smoking quit attempts and vaping quit attempts were collected at all 5 time points, while the other outcomes were collected at baseline and at 12-months. The predictors/covariates were collected either at baseline or the 12-month follow-up (detailed information are given in the Materials and Methods section. That’s why we used the word ‘mainly’. Though, to make this clear we amended this sentence ‘‘This study uses data collected over 12-months to investigate…’ line 87, page2. 

Methods: It is unclear why the description of the sample in section 2.2 and the first Table includes those who were not smokers or vapers at baseline since this is indicated as the eligibility criteria for the study.  Did you follow nonsmokers at baseline and see if anyone started smoking/vaping?

This study focused on the quitting behaviour and views, and not initiation. Therefore, only HEBECO participants who were smokers or vapers at baseline were included in the study. Separate research study will need to investigate initiation behaviours. We added in the Discussion ‘to examine how COVID-19 may stimulate interest in reducing or quitting smoking and vaping as well as initiation of these behaviours’ (lines 461-462, page15).

Table 1 provides information only for the 751 participants who were smokers and vapers at baseline. It does not include information of non-smokers and non-vapers. It is stated in section 2.2. lines 120-124, page 3, and we now added, in bold: ‘A total of 2994 participants completed the baseline questionnaire of the HEBECO study, of whom 751 were smokers and vapers (556 smokers; 337 vapers; among the smokers and vapers there were 142 dual users), who were potentially eligible for this study. Of these eligible 751 participants at baseline, 332 (44.2%) were successfully followed-up at 12 months.’

We only mention the total number of participants recruited in HEBECO study in this section as we believe this is important for the readers to understand the context of the data collection.